# Atomic structures of a bacteriocin targeting Gram-positive bacteria

Xiaoying Cai [1,2], Yao He [1,2], Iris Yu[1,2], Anthony Imani[1,2], Dean Scholl[3], Jeff F. Miller[1,2] ✉ & Z. Hong Zhou [1,2] ✉

Due to envelope differences between Gram-positive and Gram-negative bacteria, engineering precision bactericidal contractile nanomachines requires atomic-level understanding of their structures; however, only those killing Gram-negative bacteria are currently known. Here, we report the atomic structures of an engineered diffocin, a contractile syringe-like molecular machine that kills the Gram-positive bacterium *Clostridioides difficile*. Captured in one pre-contraction and two post-contraction states, each structure fashions six proteins in the bacteria-targeting baseplate, two proteins in the energy-storing trunk, and a collar linking the sheath with the membrane-penetrating tube. Compared to contractile machines targeting Gram-negative bacteria, major differences reside in the baseplate and contraction magnitude, consistent with target envelope differences. The multifunctional hub-hydrolase protein connects the tube and baseplate and is positioned to degrade peptidoglycan during penetration. The full-length tape measure protein forms a coiled-coil helix bundle homotrimer spanning the entire diffocin. Our study offers mechanical insights and principles for designing potent protein-based precision antibiotics.

*C lostridioides difficile* (*C. difficile*) is a Gram-positive pathobiont and one of the most prominent sources of nosocomial infection, responsible for almost a quarter million hospitalizations and thirteen thousand deaths per year in the US alone[1]. Dysbiosis of the gut microbiota following antibiotic exposure can disrupt colonization resistance, leading to *C. difficile* infection (CDI), which can manifest as life-threatening colitis. Treatment of CDI with antibiotics can result in further disruption of the gut microbiota and, in some cases, antibiotic-refractory recurrent disease[2]. Engineered contractile nanomachines based on R-type bacteriocins, such as the R-type diffocins of *C. difficile*[3–5] and R-type pyocins of *Pseudomonas aeruginosa*[6], as well as non-contractile nanomachines based on F-type bacteriocins, such as the F-type pyocins of *Pseudomonas aeruginosa*[7,8] and F-type monocins from *Listeria monocytogenes*[7], hold promise for developing precision medicines that kill antibiotic-resistant pathogens without harming beneficial microbes and without selecting for horizontal transfer of

resistance determinants. These bacteriocins kill by dissipating transmembrane ion gradients needed to sustain metabolic activity of their target bacteria[7].

R-type pyocins, contractile bacteriocins produced from Gram-negative bacteria, have been well studied and the structure of an R-type pyocin is described at the atomic level[9]. They resemble Type VI secretion systems (T6SS)[10], virulence cassettes from *Photorhabdus asymbiotica* (PVCs)[11], and contractile phage tails[12], as they convert chemical energy stored in the double-layered trunk region to mechanical force required to pierce target cell envelopes. Phages with a contractile tail have long been a model for studying extracellular contractile injection systems (CISs)[13]. Their effective means of binding to host bacteria and establishing a genome translocation channel through the bacterial envelope have allowed phages to successfully infect and replicate in host bacteria. As extracellular contractile machineries released by some bacteria to kill competing strains,

[1]Department of Microbiology, Immunology and Molecular Genetics, University of California, Los Angeles (UCLA), Los Angeles, CA, USA. [2]The California NanoSystems Institute (CNSI), University of California, Los Angeles (UCLA), Los Angeles, CA, USA. [3]Pylum Biosciences, San Francisco, CA 94080, USA. ✉e-mail: jfmiller@ucla.edu; Hong.Zhou@UCLA.edu

bacteriocins have similar but simplified biological constructs compared to phages[7,14,15]: they lack the DNA-containing head but still possess a needle-like central spike, a baseplate with fibers, a sheath-tube trunk, and a collar at the end of the trunk. The cylindrical trunk, comprised of a hollow tube in the center and a sheath enclosing the tube, is assembled by multiple copies of the tube and sheath proteins. Specific attachment of baseplate tail fibers to receptors on the surface of a target bacterial cell triggers conformational changes of the neighboring baseplate, which leads to reorientation of sheath proteins and contraction of the entire sheath assembly. Since the sheath and the tube are anchored together at the opposite end by the collar, collapsing the sheath pushes the tube through the baseplate, driving it to puncture the cell wall and underlying cytoplasmic membrane. As a result, the ion gradient across the membrane is breached, killing the bacterium. This highly specific (ligand-receptor driven) and efficient (mechanical penetration-based) mechanism presents great potential for applications that require precise ablation of bacterial species or strains[7].

Contractile systems that target Gram-positive bacteria, with cell walls ranging 30–100 nm in thickness, confront the challenge of penetrating a significantly thicker cell wall compared to those targeting Gram-negative organisms, which typically have cell walls only a few nanometers thick[16]. Differences in penetration mechanisms required

to puncture these two types of bacteria have been recognized[17]. For example, a peptidoglycan hydrolase/lysin is often present in Gram-positive systems (in contrast to its absence in Gram-negative-targeting systems)[18]. Though this hydrolase is conserved among many phages and its structure has been partially resolved, there is limited knowledge regarding its precise position in the contractile apparatus and its method of infiltrating the cell wall.

Here, we present the atomic structures of a diffocin, thus filling the knowledge gap regarding structures of Gram-positive-targeting contractile injection systems. The structures of the entire assembly in the pre- and post-contraction states reach 2.2 and 3.6 Å resolution, respectively; all structures were resolved using cryogenic electron microscopy (cryoEM). Comparisons between our models and other existing contractile nanomachines unveil penetration mechanisms specific to Gram-positive bacteria, as well as those shared with phages and phage tail-like nanomachines targeting Gram-negative bacteria.

## Results

### Overall structures of the diffocin in three distinct states

The diffocin gene cluster (Fig. 1a) engineered to eradicate epidemic *C. difficile* BI/NAP1/027 strains[5] was expressed in *Bacillus subtilis*, and the diffocin sample used for cryoEM imaging was purified by density gradient centrifugation. Both pre- and post-contraction diffocin

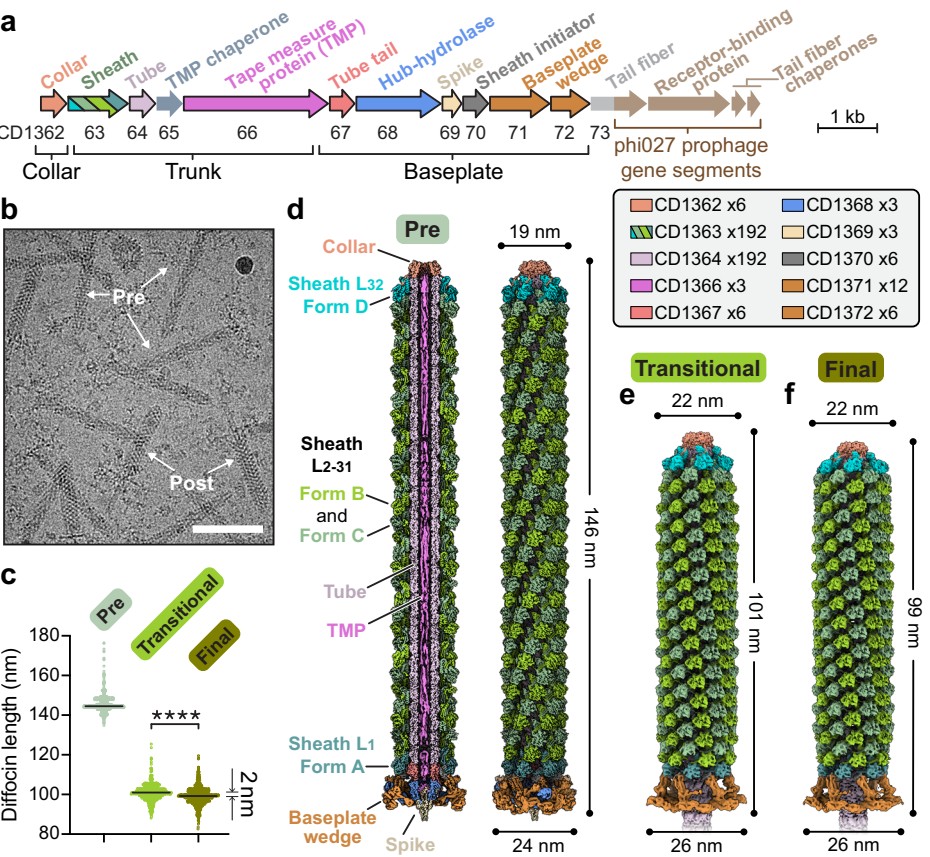

**Fig. 1 | CryoEM reconstructions of the diffocin in pre- and post-contraction states. a** Organization of engineered diffocin genes. Gene accession numbers of wild-type diffocin are shown below the corresponding genes. A portion of the gene segment encoding the tail fiber and the genes encoding the receptor-binding protein and two tail fiber chaperones were replaced with those of the phi027 prophage, which targets the epidemic *C. difficile* BI/NAP1/027 strain[5]. Genes framed in black encode proteins that are resolved in the cryoEM reconstructions. **b** A representative cryoEM image showing diffocin particles in the pre- and post-contraction states. Scale bar, 100 nm. **c** Length of diffocin particles in pre-contraction, post-contraction transitional, and post-contraction final states measured

from collar to baseplate (detailed in "Methods"). The sample sizes are 1088, 742, and 872 for the particles in the pre-contraction, post-contraction transitional, and post-contraction final states, respectively. Medians shown as black lines. Statistics performed by two-tailed unpaired *t*-test; *P* value is $1.4 \times 10^{-15}$ (****$P < 0.0001$). Source data of the length of diffocin particles are provided as a Source Data file.
**d**–**f** Composite cryoEM density maps of the diffocin in the pre-contraction (**d**), post-contraction transitional (**e**), and post-contraction final (**f**) states. A sectional view of the pre-contraction state is presented in (**d**). Structural subunits are colored as in (**a**). $L_{1-32}$ denotes layers of the sheath.

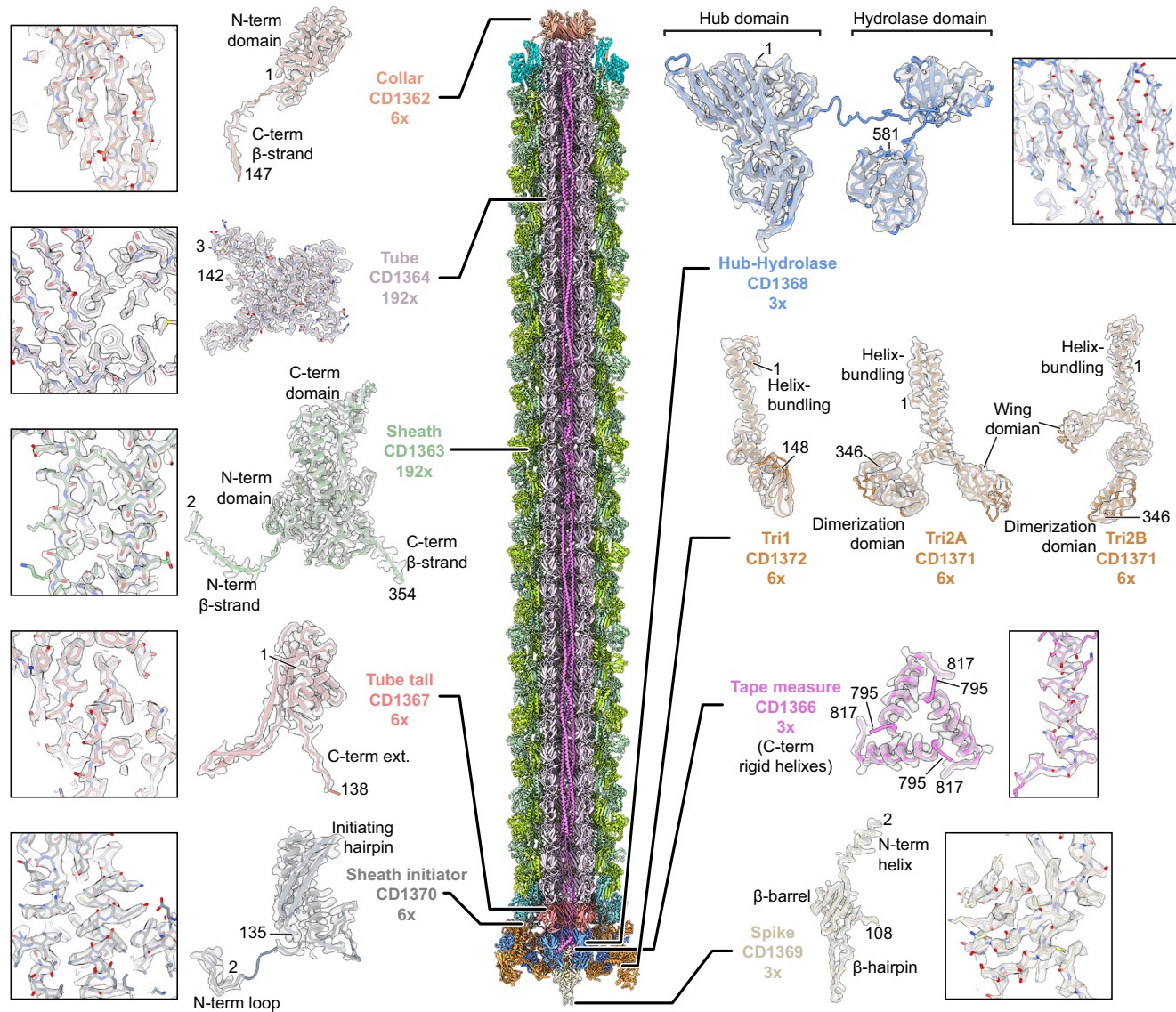

**Fig. 2 | Overview of the atomic model of diffocin in the pre-contracted state.** CryoEM densities encasing the corresponding atomic models of individual diffocin proteins in the pre-contraction state are shown on both sides of diffocin complex. Regions of the cryo-EM density map (semi-transparent densities) superimposed with atomic models (sticks) are shown in boxes, demonstrating the agreement between the observed and modeled amino acid side chains. Numbers denote chain termini.

particles were present in the purified sample (Fig. 1b). The shape of the diffocin in the pre-contraction state mirrors a syringe, and its typical length is about 146 nm (Fig. 1b–d). During contraction, the sheath layers of the trunk are compressed by about 47 nm (Fig. 1c, d, f).

There is a symmetry mismatch among the three major parts of the diffocin: the collar (C6 symmetry), the trunk (C6+helical symmetry), and the baseplate (C6 + C3 symmetry). By applying the corresponding symmetry during single-particle reconstructions, we determined cryoEM structures of the collar, trunk, and baseplate regions of pre-contraction diffocin separately at resolutions of 2.7 Å, 2.2 Å, and 2.6 Å, respectively (Supplementary Figs. 1 and 2 and Supplementary Table 1). A montaged map containing 32 layers of sheath-tube ($L_{1-32}$) was made by computationally stitching the three parts together (Fig. 1d and Supplementary Movie 1). Asymmetric densities inside the tube were further resolved by segmented symmetry relaxation. With these maps, we were able to assign 10 of the 15 diffocin gene products (Fig. 1a), including the full-length tape measure protein, and built the atomic model of the diffocin in the pre-contraction state (Fig. 2, Supplementary Fig. 3 and Supplementary Movie 2).

CIS contraction initiates from a conformational change of the baseplate, then propagates through the length of the sheath to the collar in a proposed wave-like fashion[19–21]. CryoEM data processing of the post-contraction diffocin particles resolved two different substates (Supplementary Fig. 1a, d, e and Supplementary Movie 3), an apparent transitional state and a final state, which is 2 nm shorter in length (Fig. 1c, e, f). The primary structural distinction between the two states lies in the last four sheath layers ($L_{29-32}$) adjacent to the collar region. In the transitional state these layers are only partially contracted, while they are fully contracted in the final state (Fig. 1e, f) (see "Partially contracted collar-proximal sheath layers" section below for details). The rest of the sheath layers ($L_{1-28}$) are all fully contracted in both the transitional and final states. Using the same strategy, we determined the cryoEM structures of the collar, trunk, and baseplate in the transitional and final states at 3.6–6.1 Å resolution (Supplementary Figs. 1 and 2, and Supplementary Table 1). The observation that sheath layers on the baseplate side are fully contracted, while those on the collar side are partially contracted, supports the wave-like propagation model from the baseplate to the collar, and our structure of the

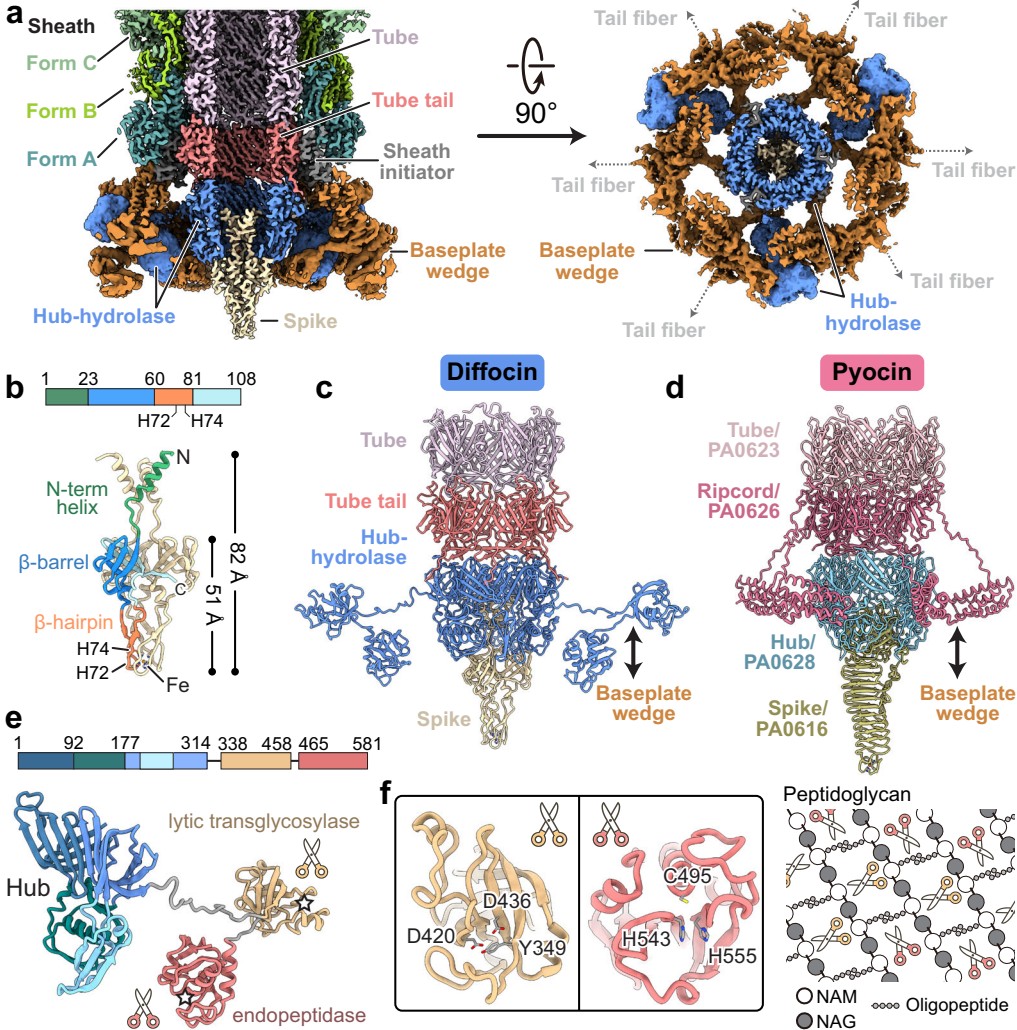

**Fig. 3 | Molecular organization of the diffocin baseplate. a** Longitudinal (left) and transversal (right) cut views of the cryoEM map of the diffocin baseplate in the pre-contraction state. **b** Ribbon diagram of the spike trimer. A monomer is colored according to the linear diagram. At the tip of the spike, His72 and His74 from three monomers collectively chelate with an iron ion. Comparison of the central parts of the diffocin (**c**) and R-type pyocin (**d**) baseplates. **e** Linear schematic and ribbon diagram of the hub-hydrolase. The hub (blue), lytic transglycosylase (yellow) and endopeptidase (red) are connected sequentially through linkers (gray). The catalytic centers of two hydrolases are labeled with pentagrams. **f** Zoom-in views of the catalytic triad of hydrolases. The lytic transglycosylase and endopeptidase cleave glycosidic bonds and peptide bonds of the peptidoglycan mesh, respectively. NAG N-acetylglucosamine, NAM N-acetylmuramic acid.

diffocin in the transitional state provides atomic resolution details of a intermediate of potential relevance to other Gram-positive CISs.

## A specialized baseplate for targeting Gram-positive bacteria

The diffocin baseplate has an inner section that connects to the tube and an outer section that connects to the sheath. The inner section comprises three proteins: tube tail (CD1367), hub-hydrolase (CD1368), and spike (CD1369) (Fig. 3a). Six copies of the tube tail form a hexameric ring immediately below the first ring of the tube. Three copies of the hub-hydrolase sit below the tube tail, enclosing the N-terminal α-helices of the spike trimer (Fig. 3a, b). The outer section of the baseplate consists of sheath initiator (CD1370) and triplex proteins (CD1371 and CD1372). Two conformers of CD1371, Tri2A and Tri2B, and one of CD1372, Tri1, form the triplex, and six copies of the triplex form the baseplate wedge, surrounding the inner baseplate (Fig. 3a and Supplementary Fig. 4a, d). The sheath initiator acts as an intermediate between the triplex and the first layer of the sheath, and between the tube tail and the hub-hydrolase (Figs. 3a and 4). While sharing a similar overall architecture, the

baseplate of the diffocin has one less protein than that of the R-type pyocin[9]. Furthermore, the diffocin baseplate has a specialized central spike and a multifunctional hub-hydrolase protein for penetrating the envelope of Gram-positive bacteria.

The central spike of the diffocin baseplate consists of three copies of the spike protein (CD1369) arranged at the end of the tube with C3 symmetry (Fig. 3b). Each spike protein has three domains: an N-terminal α-helix (residues 1–23), a conserved β-barrel with oligonucleotide/oligosaccharide-binding (OB) fold (residues 24–60 and 81–108), and a β-hairpin (residues 61–80) that is integrated into the β-barrel (Fig. 3b). Three β-hairpins collectively form the tip of the central spike. This is a distinctive feature of the diffocin as compared to other structurally characterized CISs[9,11,12,22–24], which mainly use a C-terminal β-helix as their tips (Fig. 3c, d). Interestingly, even though the diffocin spike has somewhat different architecture compared to R-type pyocin[9], it still carries a density corresponding to the ferric ion position at the tip. This putative ferric ion is coordinated by three sets of histidine doublets, His72 and His74, from the β-hairpins (Fig. 3b). Such interactions stabilize the tip for efficient puncturing through the Gram-

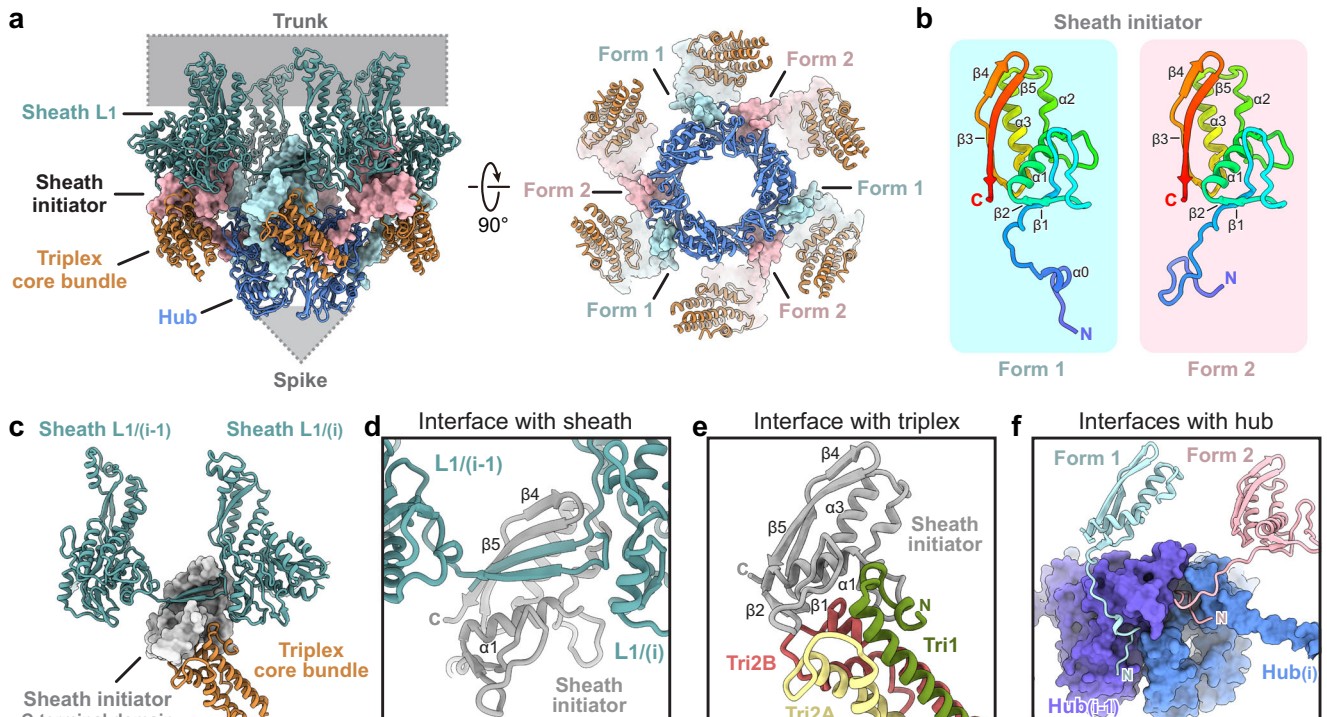

**Fig. 4 | Sheath initiator accommodates the symmetry mismatch from the sheath (C6 symmetry) to the baseplate (C3 symmetry). a** Interactions between the sheath initiator and the baseplate shown in perpendicular views. Two conformers of the sheath initiator (shown as pink and blue surfaces) bind to the hub-hydrolase alternatively. **b** Ribbon diagrams of the two conformers of the sheath initiator. The structures are rainbow colored from N-terminus (blue) to C-terminus (red). Differences between the two conformers are present at the N-terminal loops; C-terminal globular domains are the same. **c** Two interfaces on the C-terminal domain of the sheath initiator (gray surface) with the sheath and the triplex core bundle. **d** Close-up view of the handshake β-sheet formed by the C-terminal domain of the sheath initiator and two neighboring sheath subunits. **e** Close-up view of the interface between the C-terminal domain of the sheath initiator and the triplex core bundle. **f** Close-up view of the interfaces between sheath initiators and the hub-hydrolase. The N-terminal loops of adjacent sheath initiators adopt different conformations to bind at the junction of hub domains.

positive bacteria envelope while allowing the hub-hydrolase protein to degrade the peptidoglycan layer.

The 581 residue-long multifunctional hub-hydrolase protein (CD1368) is organized into two domains, a conserved unifunctional hub domain (residues 1–314) paralleling gp27 of phage T4 and a bifunctional hydrolase domain (residues 338–581), both of which are connected by a flexible loop (residues 315-337) (Fig. 3e). Three copies of the hub domain form a barrel with C3 symmetry, which functions as a 6-to-3-fold symmetry adapter between the tube tail and the central spike (Fig. 3c). The hydrolase domain has a dumbbell shape, with its two lobes binding to the peripheral region of the baseplate wedge (Fig. 3a and Supplementary Fig. 4h). The overall organization of the hub-hydrolase trimer is similar to the ripcord protein of the R-type pyocin (Fig. 3c, d), suggesting a similar role in lowering the activation energy required to trigger contraction[9]. Given that the local resolution of the peripheral region of the baseplate is only sufficient to assign secondary structural elements (Supplementary Fig. 1c and Supplementary Fig. 5a, b, g), the model of the hydrolase domain and surrounding triplex motifs were built with the assistance of AlphaFold2[25]. The two lobes of the hydrolase domain are characterized as a lytic transglycosylase (residues 338-458) and an endopeptidase (residues 465-581), respectively, which cleave two types of covalent bonds in the peptidoglycan mesh of Gram-positive bacteria (Fig. 3e, f). The lytic transglycosylase contains a conserved catalytic triad of Tyr-Asp-Asp, responsible for cleaving the β-1,4 glycosidic linkages between N-acetylmuramic acid (NAM) and N-acetylglucosamine (NAG) of the glycan strands (Fig. 3f and Supplementary Fig. 5c, d). The endopeptidase domain belongs to the NlpC/P60 family and its conserved

catalytic triad of Cys-His-His cleaves the amide bonds between amino acids within the oligopeptide chain (Fig. 3f and Supplementary Fig. 5e,f). We did not observe the density of the hydrolase domains at the spike-proximal tube region in the post-contraction state (Supplementary Fig. 5h–j) due to the limited number of particles. However, given the conserved catalytic triads of these two hydrolase domains, we speculate that the tandem hydrolases act as a pair of scissors, efficiently degrading the thick peptidoglycan layer of *C. difficile* during the contraction-initiated drilling process (see the "Discussion" section below).

**Conformational change in the baseplate wedge initiates contraction**

The baseplate wedge of the diffocin features a hexagonal iris-like architecture (Fig. 3a), which is formed by heterotrimeric complexes called the triplex (Tri2A-Tri2B-Tri1) (Supplementary Fig. 4a, d), following the nomenclature of a similar architecture in R-type pyocins[9]. Within each triplex, the N-terminal regions of the three subunits form a core bundle and a trifurcation unit, from which the C-terminal dimerization domains of Tri2A and Tri2B extend in opposite directions to interact with neighboring triplexes (Supplementary Fig. 4a, d, e). The C-terminal region of Tri1, along with the tail fiber linked to it, was not resolved in the cryoEM maps. Notably, both Tri2A and Tri2B include a wing domain inserted between the core bundle and the trifurcation unit (Supplementary Fig. 4a), reminiscent of gp6 in phage T4[12]. These wing domains contribute to the interface with the hydrolase domain of the hub-hydrolase (Supplementary Fig. 4h).

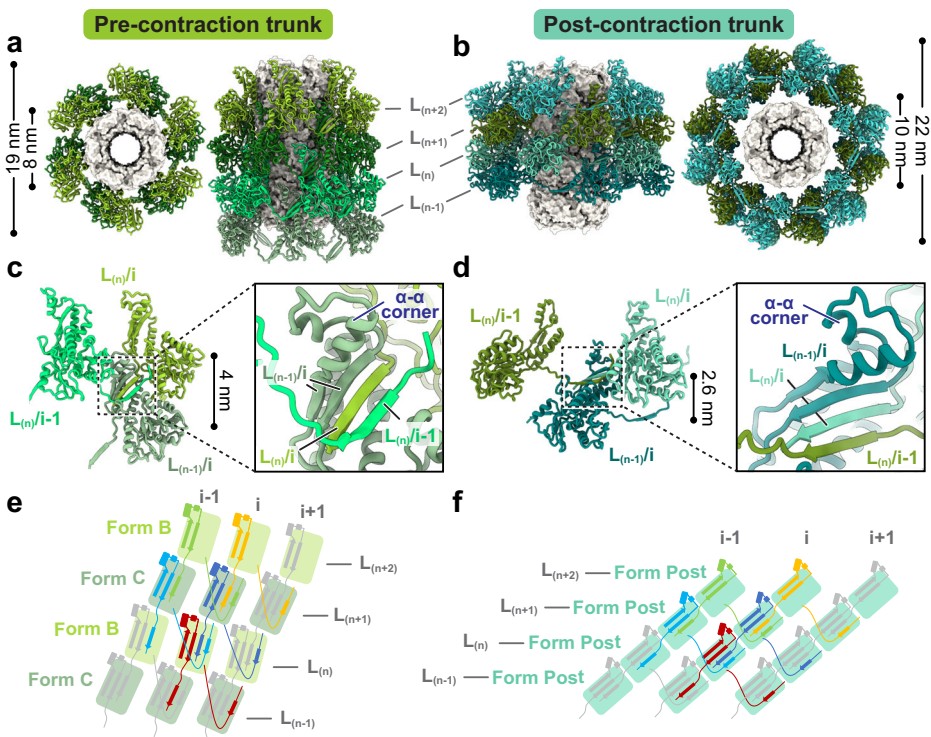

**Fig. 5 | Structure of trunk sheaths in pre- and post-contraction states.** Top and side views of diffocin trunks in the pre- (**a**) and post-contraction states (**b**). Only four layers, $L_{(n-1)/(n)/(n+1)/(n+2)}$, are shown for simplicity. Sheath proteins are presented as ribbon diagrams and tube proteins are presented as gray surfaces. The inner and outer diameters of sheath rings are labeled on the top views. Ribbon diagrams depicting interactions between neighboring sheaths in the pre- (**c**) and post-contraction states (**d**). Four β-strands from three neighboring sheath subunits jointly form the conserved handshake β-sheet. Schematic diagrams for diffocin sheath topology of the extended mesh created by the handshake interaction of augmented β-sheet in the pre- (**e**) and post-contraction states (**f**).

The baseplate wedges in the transitional and final states are almost identical, with both presenting a post-contraction conformation. By comparing them to the pre-contraction baseplate wedge, we observed repositioning of the dimerization domains of Tri2A and Tri2B, along with a twist of the trifurcation unit relative to the core bundle (Supplementary Fig. 4b, c); the dimer interface between neighboring triplexes is, however, preserved (Supplementary Fig. 4f, g). These movements result in an expansion of the iris ring from 24 to 26 nm, with the ring structure remaining intact (Supplementary Fig. 4d, f). Although the conformational change in the diffocin baseplate wedge is less dramatic than in R-type pyocin, where the iris ring breaks apart after contraction[9], it is still sufficient to initiate the contraction and release the bound hydrolase domain of the hub-hydrolase.

The contraction signal, triggered by the tail fibers attaching to receptors on the cell envelope, is transmitted from Tri1 to the baseplate wedge and then relayed to the sheath through the sheath initiator (CD1370). The N-terminal domain (residues 49–105) of the sheath initiator binds to the core bundle of the triplex, while the C-terminal β-hairpin (residues 106–135) interacts with two sheath proteins on the first sheath layer by forming a conserved handshake interface (Fig. 4c, d, e). Additionally, two conformers of the N-terminal loop (residues 1-48) from adjacent sheath initiators bind to one hub-hydrolase, intermediating the 6-to-3-fold symmetry mismatch between the baseplate wedge and hub-hydrolase (Fig. 4a, b, f).

## Trunk structures and conformational changes during contraction

The sheath protein (CD1363) has four conformations in the pre-contraction state, all sharing the same structural domains but differing in the tilt angles of the N- and C-terminal extensions (Supplementary Fig. 6a). We define these conformations as conformers A, B, C, and D. As the trunk assembles from the baseplate to the collar, conformer A forms the first layer, followed by alternating contributions from conformers B and C, and finally, conformer D terminates the sheath by interacting with the collar (Fig. 1d). Each sheath subunit engages with its neighboring sheath subunits via the conserved handshake interface, progressively assembling into the complete sheath in the form of stacked hexameric rings with helical symmetry (rise = 79.5 Å and twist = 35°) (Fig. 5a, c). The central tube inside the sheath is formed by stacked hexameric rings of tube protein (CD1364), and the interactions between the tube and different sheath conformers are the same (Supplementary Fig. 6b). The inner surface of the tube has a net neutral charge stratified between alternating positive and negative charges, similar to R-type pyocin and T6SS, but different from phages T4 and 80α (Supplementary Fig. 7).

During contraction, the diffocin sheath undergoes a transition from the alternating conformers B and C to a singular post-contraction conformer (Fig. 5e, f). The external diameter of the sheath increases from 19 nm to 22 nm, and the inner diameter increases from 8 nm to 10 nm, which enables the detachment of the sheath from the tube (Fig. 5a, b). The distance between adjacent sheath layers is decreased from 4 nm to 2.6 nm (Fig. 5c, d), resulting in a total contraction of 34%, much less than the 70% contraction of R-type pyocin[9,14]. This is evidently due to the insertion of an α-α corner motif on the tip of the C-terminal β-hairpin of the sheath protein, which acts as a spacer between adjacent sheath layers during contraction (Fig. 5c, d and Supplementary Fig. 6c). In contrast to the pre-contraction sheath, the post-contraction sheath exhibits evidently subtle motion, as revealed by the three-dimensional variability analysis of post-contraction sheath segments (Supplementary Movie 4). The helical twist and rise of these segments vary within the ranges of 27.61–28.97° and

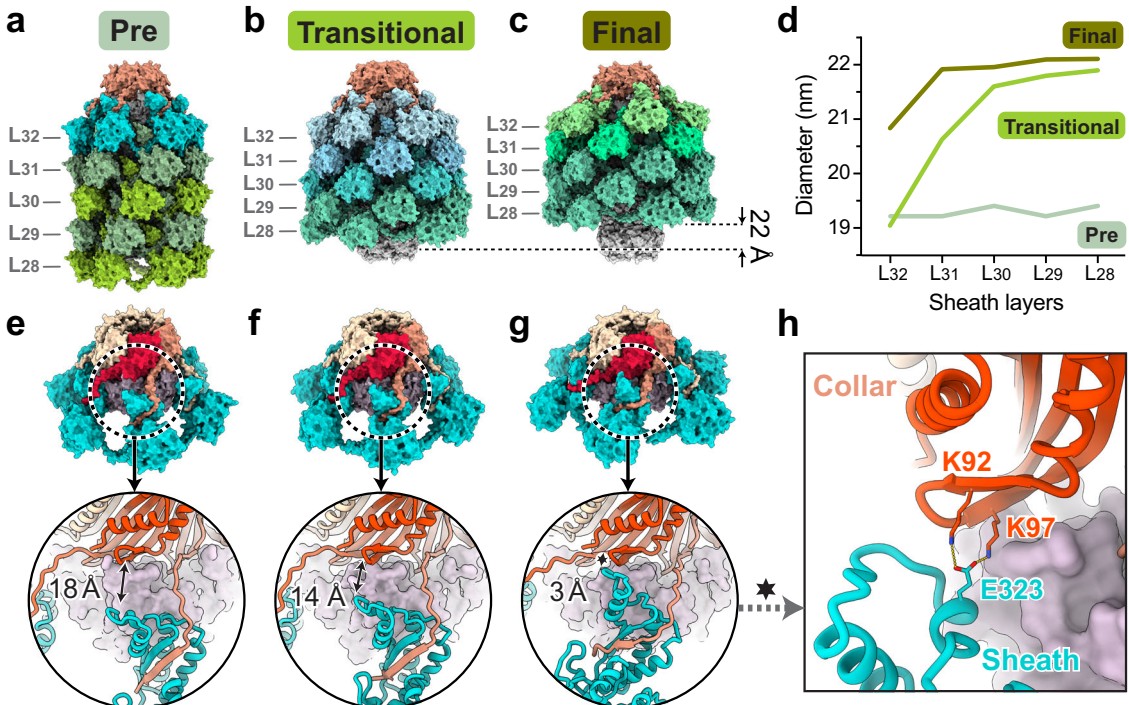

**Fig. 6 | Gradual contraction progress of collar-proximal sheaths.** Surface representations of the diffocin collars in the pre-contraction (**a**), post-contraction transitional (**b**), and post-contraction final states (**c**). The three structures are aligned using their collars. Five collar-proximal layers of the sheath ($L_{32-28}$) are shown for each structure. Sheath color is coded by different conformations. $L_{29-28}$ of sheath in the transitional state and $L_{30-28}$ of sheath in the final state are trunk sheathes with the post conformer. **d**, Plot of outer diameters of sheath $L_{32-28}$ in three states. **e**–**g** Views showing the interface between the collar and sheath layer 32 ($L_{32}$) in three states. **h** Additional interactions between collar and sheath $L_{32}$ that only appear in the final state, labeled as hexagram in right panel of (**g**).

25.36–26.72 Å, respectively (Supplementary Fig. 2c, d). This is in line with the increase of entropy from the metastable, higher energy pre-contraction state to the lowest energy post-contraction state.

## Partially contracted collar-proximal sheath layers captured in a diffocin transitional state

The inner tube and outer sheath are tethered together by a hexameric ring of collar protein (CD1362) at the collar-proximal end of the diffocin (Fig. 6a). The N-terminal domain of the collar subunit binds to the tube, while its C-terminal β-strand extends towards the C-terminal β-hairpin of last sheath, forming a three-β-strand handshake interaction (Fig. 6e). These interfaces are maintained during contraction (Fig. 6e–g), enabling the collar to function as a force transducer between sheath contraction and tube ejection.

We captured two states of the diffocin in solution after contraction (Fig. 6b, c), which show different structures in the last four sheath layers ($L_{29-32}$) adjacent to the collar region (Fig. 6a–d) and in the interfaces between the collar and last sheath layer (Fig. 6e–h). In the transitional state, these sheath layers are partially contracted, as indicated by the gradually decreasing diameter of the hexameric sheath rings from $L_{28}$ to $L_{32}$ (Fig. 6d). Specifically, the diameter of $L_{28}$ in the transitional state is similar to that in the final state, while the diameter of $L_{32}$ in the transitional state is almost identical to that in the pre-contraction state (Fig. 6d). Consequently, the last few sheath layers still interact with the central tube in the transitional state. These features closely resemble the computationally simulated transition state of the R-type pyocin sheath-tube complex[19]. The conformational change from the transitional state to the final state results in a 37° clockwise rotation and additional 2 nm injection of the central tube. In addition to the interface between the C-terminal β-strand of the collar and the C-terminal β-hairpin of the last sheath in the transitional state (Fig. 6f) a second interface between the collar and the last sheath is observed in

the final state (Fig. 6g, h). Lys92/97 on the N-terminal domain of the collar form salt bridges with Glu323 on the α-α corner motif of the last sheath (Fig. 6h). These interactions might provide extra mechanical stability to the junction after contraction. The insufficient contraction of the sheath and the absence of additional collar-sheath interactions indicate that the transitional state is metastable relative to the final state and has potential for further contraction.

## The tape measure protein forms a coiled-coil trimer

The length of phage tails and other phage tail-like injection systems is determined by a tape measure protein (TMP), which is located in the lumen of central tube[24,26–29]. To date, no structure has been reported for any full-length TMP, likely due to its intrinsic flexibility and mismatch of symmetry with the rest of the respective bacteriocin or phage tail. It has been demonstrated that TMPs assemble either as trimers or hexamers; twenty and thirty-five C-terminal residues of the TMPs of phages 80α and T5, respectively, were previously resolved as trimers[29,30], while sixfold helical features were suggested for the TMP from phage Pam3[24].

To resolve the asymmetric structure of the full-length diffocin TMP (CD1366), we divided the pre-contraction diffocin particles into six segments, spanning from the collar to the baseplate, and gradually relaxed their symmetry from C6 to C3 to C1 during cryoEM data processing using featureless spherical masks (Supplementary Fig. 8). A composite map of the full-length TMP was generated by montaging the asymmetric densities of the TMP from each segment (Fig. 7a, Supplementary Fig. 8d and Supplementary Movie 5). In the map, TMP densities run along the entire tube lumen from the collar to the baseplate, with a length of ~1388 Å and a diameter of ~25 Å (Fig. 7a, h). About 75% of the TMP densities exhibit features of three distinct rod-shaped densities around the central axis with quasi-C3 symmetry, reminiscent of a coiled-coil composed of three helixes (Fig. 7a–e). TMP

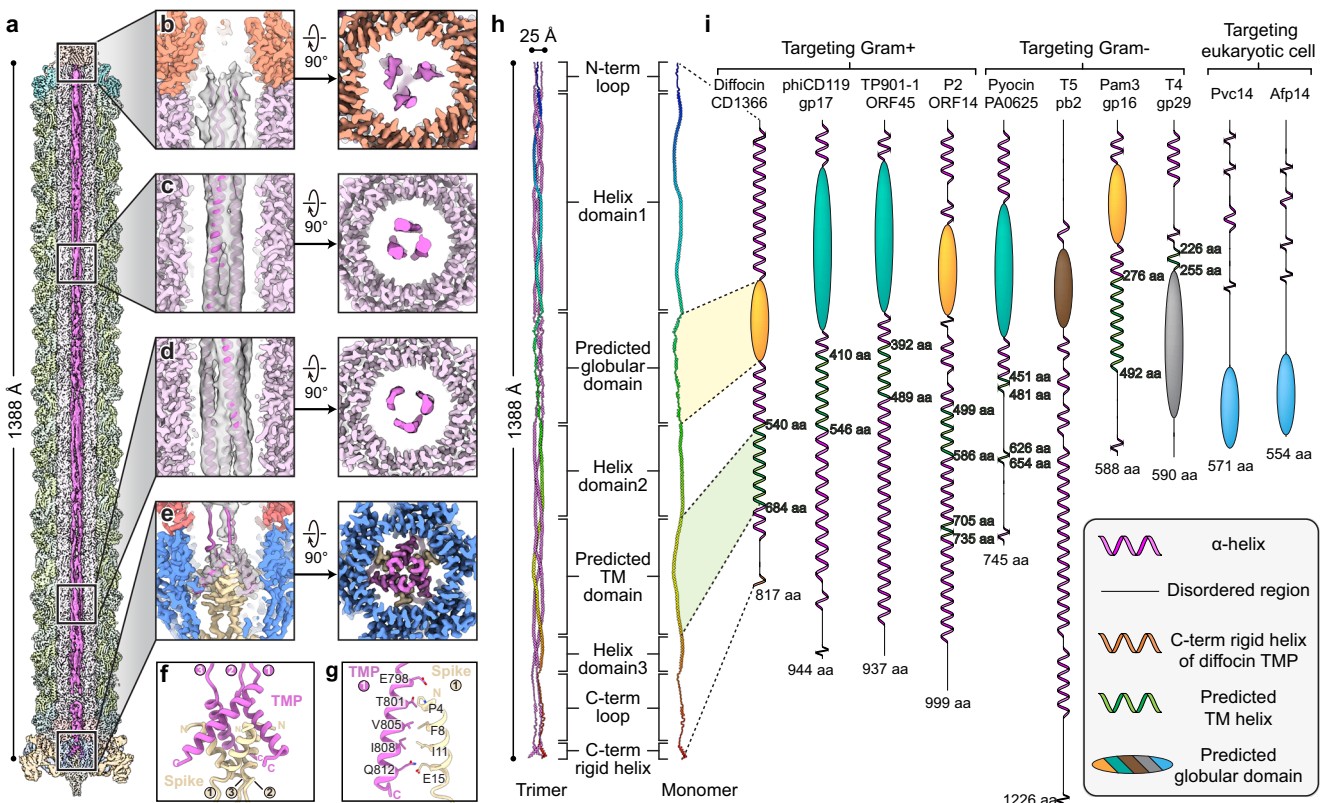

**Fig. 7 | Tape measure protein (TMP) of the diffocin. a** Sectional view of composite cryoEM density map of the diffocin in the pre-contraction state. TMPs are colored in magenta and the other components are in semitransparent colors. **b–e** Zoom-in views of diffocin TMP densities as indicated in (**a**). Corresponding top views are shown on the right. TMP models are fitted into the cryoEM densities in side views. **f** Interface between the TMP trimer and spike trimer. **g** Interactions between the C-terminal helix of the TMP and the N-terminal helix of the spike. Residues on the interface are labeled and shown in stick representation. **h** Model of the full-length diffocin TMP. One TMP subunit is rainbow colored from its N-terminus (blue) to C-terminus (red). **i** Illustration of predicted secondary structures of TMPs from phages and phage tail-like nanomachines. Alpha-helices, transmembrane (TM) domains and disordered regions were predicted by Phyre2[31]. Globular domains were predicted by AlphaFold2[25]. Globular domains in the same color indicate their structural similarity (see details in Supplementary Fig. 9).

densities in the baseplate region reveal high-resolution features with C3 symmetry, and the C-terminal α-helices (residues 795-817) of three TMPs were de novo built into the densities (Fig. 7e–g). These α-helices interact with the N-terminal α-helices of the spike trimer by forming an intertwined helix bundle (Fig. 7f, g). The rest of the TMP (residues 1-794) was modeled with the assistance of AlphaFold2[25], which predicted most of the secondary structures as α-helices. The density of the TMP ends at the collar region and it displays higher flexibility (Fig. 7a, b). In summary, the diffocin TMP assembles as a trimer with a coiled-coil pattern (Fig. 7h), with the N-termini interacting with the collar and the C-termini anchored on the spike.

In addition to determining the length of the diffocin, other potential functions of TMP were predicted by AlphaFold2[25] and Phyre2[31] structure prediction tools. AlphaFold2 predicted a tertiary structure of residues 289-425 of CD1366 with relatively high pLDDT confidence score, depicting it as a globular domain (Fig. 7h, i and Supplementary Fig. 9a). This predicted globular domain (PGD) is comprised of a series of short (<15 amino acids) α-helices connected by turns/loops (Supplementary Fig. 9a). Its corresponding cryoEM density is relatively flexible and appears as multiple short rods connected linearly (Fig. 7a), indicating that these series of short α-helices are linearly arranged in the lumen of central tube (Fig. 7a, h). Additionally, Phyre2[31] predicts that residues 540–684 of CD1366 form five transmembrane (TM) helices (Fig. 7h, i).

## Discussion

By providing an atomic description of a bactericidal contractile system against Gram-positive bacteria, the atomic structures of the diffocin

complex in its pre- and post-contraction states reported here offer a number of important insights into the mechanism of how this system works as compared to CISs targeting Gram-negative bacteria.

CIS contraction initiates at the baseplate. Comparison between the baseplate components of the diffocin and R-type pyocin complexes highlights the unique challenge that this Gram-positive-targeting CIS faces regarding thick cell wall degradation. The hydrolase domain of the hub-hydrolase of the diffocin functions like a pair of scissors made up of two blades: the lytic transglycosylase and the endopeptidase, which have conserved catalytic triads of Tyr-Asp-Asp and Cys-His-His, respectively (Fig. 3e, f and Supplementary Fig. 5a–f). These scissors are predicted to catalytically cut the glycosidic linkages of glycan strands and the peptide bonds of oligopeptide chains in the peptidoglycan mesh with the aforementioned conserved catalytic triads (Supplementary Fig. 5c–f), thereby destroying the thick peptidoglycan layer of *C. difficile*. This bifunctional hydrolase domain not only works as a hydrolase but also acts like the ripcord protein of R-type pyocin[9]. It binds tightly to the dimerization ring of the baseplate in the pre-contraction state. When the dimerization ring receives a triggering signal from the tail fiber, the conformational change of the triplex will cause the release of the hydrolase domain from the dimerization ring. As the spike drills into the Gram-positive cell, the released hydrolase domain would commence its catalytic function in a rotational motion to degrade the cell wall of the Gram-positive bacteria.

Contraction of known CISs propagates continuously through the sheath, eventually ending at the collar. However, contraction of the diffocin pauses at collar-proximal sheath layers prior to completion,

giving rise to two post-contraction states. The four-layer sheath near the collar only contracts partially in the transitional state, allowing for the tube to be extended an additional 2 nm in the final state (Fig. 1e,f and Fig. 6b,c). After contraction to the transitional state, the tube protrudes 45 nm away from the baseplate (Supplementary Fig. 10b). This protruding tube may encounter difficulty in penetrating the entire cell envelope of *C. difficile*, which is composed of an outermost S-layer, followed by a thick peptidoglycan layer and a lipid bilayer inner membrane[32] with an average thickness of approximately 58 nm[33]. Rather, it may be necessary to pause briefly, allowing the hydrolase domain of the hub-hydrolase to continue its catalytic action, thoroughly degrading the thick peptidoglycan layer of *C. difficile*. The stored chemical energy remaining in the transitional state would be used in the final contraction to eject the TMP from the tail tube lumen. In the absence of spatial constraints, the coiled-coil TMP of the diffocin will likely refold to adopt a globular domain with five TM helices once detached from the collar and spike and released from the tube lumen during contraction. As a result, fifteen TM helices can be inserted and form a sizeable pore on the inner membrane of the envelope, dissipating the cellular electrochemical gradient and killing the bacterium (Supplementary Fig. 10c). This step-wise contraction mechanism may be used to accommodate the characteristics of the envelope structure of Gram-positive bacteria.

In addition to their primary function of determining the length of the phage tail, phage TMPs are involved in a myriad of other secondary, host-related functions after ejection from the tail tube lumen: particular amino acid sequences were identified as peptidases responsible for degrading the host cell wall[30,34,35], RNA polymerases used for transcribing DNA[36], and TM domains facilitating the formation of a channel on the inner membrane for DNA translocation into the host cell[37]. Like the diffocin TMP, the TMPs of Gram-positive-targeting phages phiCD119, TP901-1 and P2 also have a predicted TM region (Fig. 7i), with five, three, and four TM helices, respectively, potentially forming a sizeable channel that assists genomic transduction[37,38]. R-type pyocin contracts 70 nm upon injection into the cell[9,19], greater than the approximately 29 nm cell envelope of *Pseudomonas aeruginosa*[39], a Gram-negative bacteria composed of a periplasm containing a thin peptidoglycan layer sandwiched between the lipid bilayer outer and inner membranes[16,39]; therefore, its protruding tube may completely penetrate the thin peptidoglycan layer after contraction with relative ease and subsequently puncture into the inner membrane of the target cell (Supplementary Fig. 10a). Gram-negative-targeting, such as phages T4 and T5, present little to no predicted TM region within their TMPs (Fig. 7i). Cyanophage Pam3 is a notable exception, containing a 217 residue-long TM region (Fig. 7i) that can form seven TM helices. Pam3 targets the Gram-negative cyanobacteria *Pseudanabaena mucicola* that also has a thick peptidoglycan and S-layer similar to those in Gram-positive bacteria[40,41]. Its TMP may employ its TM features to form a genomic transduction channel across the inner membrane resembling Gram-positive-targeting phage TMPs. Thus, we propose that phages and phage tail-like bacteriocins targeting cells with a thick peptidoglycan layer have a prominent TM region, possibly forming a sizeable conduit on the cell inner membrane to assist with genetic transduction and cell killing, respectively.

AlphaFold2[25] predicts a globular domain within the diffocin TMP, consistent with similar PGDs present in other phages and phage tail-like nanomachines (Fig. 7i). The PGDs of diffocin CD1366, P2 ORF14 and Pam3 gp16 share a conserved tertiary structure; the PGDs of R-type pyocin PA0625, phiCD119 gp17 and TP901-1 ORF45 share a conserved tertiary structure; the PGDs of Pvc14 and Afp14 share a conserved tertiary structure (Fig. 7i and Supplementary Fig. 9b). Searching for homologous proteins with NCBI BLAST[42] or similar structures in Dali server[43] yielded no matches for any of these PGDs, suggesting a lack of known conserved function. The PGD of the Gram-positive-targeting diffocin shares structural similarities with that of phage counterparts

P2 (targeting Gram-positive bacteria) and Pam3 (targeting Gram-negative bacteria), suggesting a lack of specificity based on cellular envelope character (Fig. 7i and Supplementary Fig. 9b). PGDs of the Gram-negative-targeting R-type pyocin and the PGDs of phages phiCD119 and TP901-1, both of which target Gram-positive bacteria, have similarly predicted structures, further highlighting the non-discriminatory nature of PGDs (Fig. 7i and Supplementary Fig. 9b). Notably, all aforementioned phages and phage tail-like bacteriocins contain a PGD within their TMP that indicates potential non-selective supplementary features of TMPs upon ejection from their quasi-linear conformations in the tube.

Bacteriocins hold great promise as precision antibiotics to alleviate the widespread bacterial resistance problem facing humanity arising from decades of overusing wide-spectrum antibiotics. Looking forward, the structure of a contractile bacteriocin targeting a Gram-positive bacterium presented here opens the doors for engineering potent, nucleotide-free contractile nanomachines for applications in biomedicine and food industry. The diffocin construct we imaged has a modified receptor-binding domain that specifically targets the *C. difficile* S-layer protein so that it kills epidemic lineages of the bacteria in the gastrointestinal tract[4,5]. To target other Gram-positive bacterial pathogens, the TMP and the receptor-binding domain should be changed to match the characteristics of the specific pathogens according to the thickness of their cell envelope and surface receptor. Future computational and cellular cryogenic electron tomography studies similar to those applied to R-type pyocin[19] and phage infection[44] should shed light on other transient structural states and mechanism of action of the diffocin-mediated killing of *C. difficile* cells, allowing for engineered optimization.

## Methods
### Purification of diffocins for cryoEM
The diffocin construct used in this study was Av-CD291.2/construct KX592438, which was engineered to eradicate *C. difficile*, especially the BI/NAP1/027 strain type[5]. This construct has a modified receptor-binding domain that specifically targets the *C. difficile* S-layer protein. For production, the diffocin gene cluster was integrated into *B. subtilis* genome and was induced as previously described[5]. Diffocins were purified from crude lysates by polyethylene glycol precipitation followed by differential centrifugation with a final ultracentrifuge step to pellet diffocin particles. This sample was further purified using a 10% to 50% sucrose (w/v) gradient at 77,000 × *g* for 1.5 h at 4 °C. After centrifugation, one band was visible at about 25% position and was extracted gently by fractionation with a 100 μL pipette from the top of the centrifuge tube along its side. The extracted sample was then diluted to a final volume of 4 mL with Tris buffer (10 mM Tris, 130 mM NaCl, pH 7.4). The diluted sample was concentrated using a 100 kDa Amicon molecular filter to approximately 50 μL. This dilution-concentration step was repeated 3 more times in the same filter to remove the gradient material, resulting in a final sample volume of 50 μL in Tris buffer for cryoEM imaging.

### CryoEM data collection
For cryoEM data collection, a 2.5 μL aliquot of the purified diffocin sample was loaded onto a Quantifoil® 1.2/1.3, 200 mesh grid, blotted for 5 seconds at force 2, and then flash-frozen in liquid ethane with a Vitrobot Mark IV (FEI/Thermo-Fisher). CryoEM grids were loaded into a Thermo Fisher Titan Krios electron microscope operated at 300 kV for automated data collection using SerialEM[45]. Movies of dose-fractionated frames were acquired with a Gatan K3 direct electron detector in super-resolution mode at a pixel size of 0.55 Å on the sample level. The total dose rate on the sample was set to ~50 electrons per Å², which was fractionated into 50 frames with an exposure time of 0.06 s for each frame. A total of 29,750 movies were acquired in two imaging sessions. Frames within each movie were 2× binned (pixel size

of 1.1 Å), aligned to correct beam-induced drift, and dose weighted using UCSF MotionCor2[46]. Dose-weighted micrographs were used for the following CTF determination, particle picking and final reconstruction. Contrast transfer function (CTF) parameters of each micrograph were determined by CTFFIND4[47].

## CryoEM data processing

CryoEM data processing workflows are outlined in Supplementary Fig. 1 and 2. All steps described below were performed with RELION 4.0[48] unless otherwise indicated. Resolutions of the cryoEM maps were estimated on the basis of the gold standard[49] Fourier shell correlation (FSC) = 0.143 criterion. Local resolution evaluations were determined by RELION with two independently refined half-maps. Data collection and processing statistics are given in Supplementary Table 1.

Diffocin particles in both the pre- and post-contraction states were present in cryoEM micrographs and can be readily distinguished by eye (Fig. 1b). The two ends (collar and baseplate) of each diffocin were manually picked from 2000 representative micrographs and screened by 2D classification. Particles from the best classes were selected to train a particle detection model in Topaz[50] for subsequent neural network-based particle picking from all micrographs. The picked particles were extracted in dimensions of 300 × 300 square pixels for both the pre- and post-contraction states. After several rounds of 2D classification and 3D classification with C6 symmetry, 202,971 collar particles in the pre-contraction state (pre-collar), 414,022 baseplate particles in the pre-contraction state (pre-baseplate), 117,846 collar particles in the post-contraction state (post-collar), and 30,712 baseplate particles in the post-contraction state (post-baseplate) were selected and processed separately in the following steps (Supplementary Fig. 1a).

For the pre-collar, the particles were shifted along the C6 axis if necessary to ensure that their collars were at the same height. After an additional round of 3D classification, 144,368 pre-collar particles from the best class were selected and refined to 2.7 Å resolution with C6 symmetry.

For the pre-baseplate, 353,620 particles were selected after 3D classification and refined to 2.6 Å resolution with C6 symmetry. However, densities at the spike region were much worse than those at the trunk region, suggesting there was a symmetry mismatch between the spike and the trunk as observed in other CISs. After relaxing symmetry from C6 to C3, we obtained a 2.7 Å resolution reconstruction of the pre-baseplate with reasonable densities at the spike region. To improve the density of the baseplate triplex, 116,539 particles were selected after an alignment-free 3D classification focused on the triplex region and refined to 2.9 Å resolution with C3 symmetry imposed.

For the post-collar, the initially selected 117,846 particles were a mixture of post-collar and post-trunk particles. To separate them, we conducted an alignment-free 3D classification using a spherical mask covering the collar region. Additionally, two types of post-collars were separated: one is more compressed (shorter) than the other one (Supplementary Fig. 1a). After removing bad particles, we obtained C6-symmetrized reconstructions of the shorter collar (post-collar short) at 3.9 Å resolution and the longer collar (post-collar long) at 3.6 Å resolution (Supplementary Fig. 1a).

For the post-baseplate, 24,004 particles were selected after 3D classification and refined to 5 Å resolution with C6 symmetry. These particles were further classified into two groups based on the relative location between the inner tube and the baseplate. The two groups of particles were refined separately with C6 symmetry to 5.6 Å resolution (post-baseplate state1) and 6.1 Å resolution (post-baseplate state2) (Supplementary Fig. 1a). We believe that post-baseplate state1 and post-baseplate state2 correspond to post-collar long and post-collar short, respectively; this is because the offset of the tube between post-baseplate state1 and post-baseplate state2 matches with the offset of collar between post-collar long and post-collar short sets. This is

confirmed by manual inspection of these particles on cryoEM images (see "Length statistics of diffocin in different contraction states" section for details). Post-collar long is the collar of diffocin in the transitional state and post-collar short is the collar of diffocin in the final state, as illustrated in Fig. 6. Thus, post-collar long, post-collar short, post-baseplate state1 and post-baseplate state2 were defined as post-collar transitional, post-collar final, post-baseplate transitional and post-baseplate final, respectively (Supplementary Fig. 1d,e).

For the trunk regions, the helical rise of the pre- and post-contraction states were initially determined to be 40 Å and 25 Å, respectively, based on the reconstructions of the collar and baseplate (Supplementary Fig. 1). Trunk segments separated during the initial 3D classification step (Supplementary Fig. 1a) were used as "seeds" for particle picking. These trunk segments (G0 segment) were refined with C6 symmetry. The resulting center of each segment was shifted along the helical axis (Z-axis) in both directions by one layer to generate new segments. These segments were combined with G0 segments and re-extracted after removing duplicates (inter-segment distance less than 1.5 layers). These newly extracted segments were refined with C6 symmetry, and then subjected to alignment-free 3D classification to remove poor particles. The resulting good segments (G1 segments) were shifted by another layer to generate G2 segments following the same procedures. After 3–4 rounds of shifting and selection, 2.3 million pre-trunk segments and 1.1 million post-trunk segments were extracted in dimensions of 300 × 300 square pixels (Supplementary Fig. 2a), with each segment having at least one unique asymmetric unit. The extracted segments were subjected first to 2D classification, followed by a 3D classification to eliminate poor particles. The selected pre-trunk segments were 3D refined with C6 symmetry and helical symmetry (helical rise of 40.0 Å and helical twist of 17.5°). The following 3D classifications yielded two distinct classes that differed only by a shift of ~40 Å along the helical axis and a rotation of ~17.5°. We shifted the segments in one class by 40 Å and then combined them with the segments in the other class. Duplicated segments were removed (inter-segment distance less than 100 Å). Finally, 450,732 pre-trunk segments were refined to 2.2 Å resolution with C6 and helical symmetry imposed. The refined helical rise is 79.5 Å and the refined helical twist is 35.0° (Supplementary Fig. 2a). For the post-trunk, continuous heterogeneity of sheath layer was observed and analyzed by cryoSPARC 3D Variability Analysis (3DVA)[51] (see "Quantitative analysis of conformational heterogeneity of post-trunk" section for details). Finally, 65,376 post-trunk segments selected from alignment-free 3D classification were refined to 3.6 Å resolution with C6 and helical symmetry imposed, and the refined helical rise and twist of this reconstruction were 25.6 Å and 27.8°, respectively.

For the tape measure protein, each pre-contraction diffocin particle was boxed into six segments (one collar segment, four trunk segments, and one baseplate segment) for processing (Supplementary Fig. 8a, b). The box size of these segments was 300 pixels, and the center-to-center distance between neighboring segments was 250 pixels. The pre-collar, pre-trunk 1, and pre trunk 2 segments were initially refined with C6 symmetry and the pre-trunk 3, pre-trunk 4, and pre-baseplate segments were refined with C3 symmetry (Supplementary Fig. 8a). The C6 symmetry segments were subsequently relaxed from C6 to C3 by alignment-free 3D classifications using spherical masks covering the tape measure protein regions. For each type of segment, three 3D classifications were conducted in parallel with three spherical masks (diameter of 110 pixels) distributing along the Z-axis. The resulting classes with good tape measure protein features were selected, and then relaxed symmetry from C3 to C1 by another round of alignment-free 3D classifications using the same masks. The classes with good features of the tape measure protein were selected (Supplementary Fig. 8c). The C3 symmetry segments were subsequently relaxed from C3 to C1 symmetry via a near identical process, with the only difference being the use of a 60-pixel

diameter mask for the pre-baseplate segment, as opposed to the aforementioned 110-pixel mask used in all other segments (Supplementary Fig. 8c). Finally, a composite map of entire tape measure protein was generated by montaging 20 pieces of small maps from six segments (Supplementary Fig. 8d).

### Model building and refinement

Both the pre-contraction and post-contraction atomic models were built in Coot[52]. CD1362, CD1363, CD1364, CD1366 (residues 795-817), CD1367, CD1368 (residues 1-314), CD1369, CD1370, CD1371 (residues 1-76) and CD1372 (residues 1-80) were modeled de novo. With cryoEM density maps of the pre-contraction state at 2.2–2.9 Å resolution, we were able to clearly and confidently assign not only the backbone α-carbon positions, but also side chain identities. For each protein chain, we identified regions with strong secondary structures, such as α-helices, allowing us to determine the peptide direction during initial modeling stages. We placed batons to locate the position of α-carbons within the density map and converted the batons into poly-alanine chains with the mainchain function. The poly-alanine chains were then mutated into its proper alignment sequence. Large protruding side chains in the density map belonging to phenylalanine, tyrosine, and tryptophan residues were used as markers to align the sequence to the map. De novo modeling was not applied to proteins CD1366 (residues 1-794), CD1368 (residues 315-581), CD1371 (residues 77-346) and CD1372 (residues 81-148) due to their weak densities in the map. Predicted tertiary structures for these proteins were generated by AlphaFold2[25]. We fitted the predicted structures into the density and refined manually in Coot[52]. For cryoEM density maps of the post-contraction state at 3.6–6.1 Å resolution, we used the pre-construction models as initial models. These initial models were docked into the density maps of post-contraction diffocin by rigid-body fitting, followed by manually rebuilding in Coot[52].

Atomic models were refined through an iterative process between automatic refinement with Phenix[53] and manual corrections in Coot[52]. For the automatic refinement step, the atomic models were refined using the *phenix.real_space_refine* command of the Phenix package[53] with default refinement parameters. First, we refined individual proteins with the whole cryoEM maps to improve their secondary structure, Ramachandran, rotamer restraints, and intramolecular clashing score. Then, we combined all proteins to generate the full models of the baseplate, collar and trunk in Coot[52] and refined the full models in Phenix in order to separate clashing atoms between adjacent monomers. At each of the refinement steps, we manually inspected the models to assess quality of the refinement, made manual adjustments and repeated the refinement steps until a final structure was reached.

The final refined models were validated with EMRINGER Score[54], Ramachandran Plot, C-beta, and map CC as well as MolProbity[55] and the results are summarized in Supplementary Table 1. Figures and movies were generated with UCSF ChimeraX[56].

### Length statistics of diffocin in different contraction states

The collar and baseplate particles used for the final cryoEM reconstructions were mapped back to raw cryoEM images, and then manually inspected to match up the collar and baseplate from the same diffocin particle. Diffocin particles presenting only the collar or baseplate or those with ambiguous features were excluded from consideration. Through the manual inspection, we also confirmed that post-baseplate state1 and post-baseplate state2 correspond to post-collar long and post-collar short, respectively (Supplementary Fig. 1a). Finally, 1088 collar-baseplate pairs from pre-contraction diffocin, 742 pairs from post-contraction transitional state diffocin, and 872 pairs from post-contraction final state diffocin were selected from 1000 cryoEM images for analysis. For each collar-baseplate pair, the distance from the center of the collar to the center of the baseplate was

calculated using their coordinates. The length of each diffocin particle was calculated by adding up the distance from the center of the collar to the center of the baseplate, the distance from the center of the collar to the top of the collar (measured from the cryoEM map of collar), and the distance from the center of the baseplate to the bottom of the baseplate (measured from the cryoEM map of baseplate).

### Quantitative analysis of conformational heterogeneity of post-trunk

528,409 post-trunk segments initially refined with C6 and helical symmetry were subjected to 3DVA[51] in cryoSPARC and separated into 20 clusters based on the 3DVA result (Supplementary Fig. 2c). Segments in each cluster were refined individually with C6 and helical symmetry imposed (Supplementary Fig. 2d). Helical rise and twist for each cluster were refined as well.

### Reporting summary

Further information on research design is available in the Nature Portfolio Reporting Summary linked to this article.

## Data availability

The cryoEM density maps and corresponding atomic models have been deposited in the EMDB and PDB, respectively. The accession numbers are listed as follows: pre-contraction collar (PDB: 8V3T and EMD-42953); post-contraction collar in transitional state (PDB: 8V3Z and EMD-42961); post-contraction collar in final state (PDB: 8V40 and EMD-42962); pre-contraction baseplate reconstructed in C6 symmetry (EMD-42957); pre-contraction baseplate reconstructed in C3 symmetry (EMD-42958); pre-contraction baseplate focused refinement on triplex region (PDB: 8V3W and EMD-42956); post-contraction baseplate in transitional state (PDB: 8V41 and EMD-42963); post-contraction baseplate in final state (PDB: 8V43 and EMD-42964); pre-contraction trunk (PDB: 8V3X and EMD-42959); post-contraction trunk (PDB: 8V3Y and EMD-42960). Source data of the length of diffocin particles are provided with this paper. Source data are provided with this paper.

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

## Acknowledgements

We thank former members of the Zhou lab including Ke Ding, Justin Bui, Wesley Shen, and Peng Ge for initial research efforts of the project, Jane Lee and summer high-school students Daniel Doran and Julia Greenbaum for picking some of the baseplate and collar particles in the preliminary micrographs at the initial stage of the project. This research was supported in part by NIH (R01GM071940/DE028583/DE025567 to Z.H.Z. and R33AI121692/R21AI085318 to D.S.), and the Schaffer Family Foundation and Kavli Endowment (to J.F.M.). We acknowledge the use of resources at the Electron Imaging Center for Nanomachines [supported by UCLA and by instrumentation grants from the NIH (1S10OD018111) and NSF (DBI-1338135 and DMR-1548924)].

## Author contributions

Z.H.Z., J.F.M., and D.S. conceived the project; D.S. prepared the samples; X.C. and Y.H. recorded cryoEM images; X.C. and Y.H. determined the cryoEM structures; X.C., I.Y., Y.H., and A.I. built the atomic models; Z.H.Z., X.C., Y.H., I.Y., D.S. and J.F.M. interpreted the models; X.C., Y.H., A.I. and I.Y. made figures. Z.H.Z., J.F.M., X.C., Y.H., A.I., and I.Y. wrote the paper; and all authors contributed to the editing of the manuscript.

## Competing interests

J.F.M. is a cofounder, equity holder and chair of the scientific advisory board of Pylum Biosciences, Inc., a biotherapeutics company in San Francisco, CA, USA. D.S. is an employee and equity holder of the same company. The remaining authors declare no competing interests.
