## [Peer Review File · Nature Communications]

Reviewers' Comments:

Reviewer #1:

Remarks to the Author:

In this manuscript, Cai et al. reported the cryo-EM structures of the diffocin from bactericidal contractile system against Gram-positive bacteria in its pre- and post-contraction states, enriching our understanding on the contraction injection system. Furthermore, they found that the diffocin harbors a multifunctional hub-hydrolase protein that connects the tube and baseplate for the degradation of peptidoglycan during penetration. Additionally, the full-length tape measure protein forms a coiled-coil helix bundle homotrimer spanning the entire length of the diffocin. The following major concerns are raised for the authors to revise and improve the manuscript.

1) Line 264-273: In the transitional state, the four-layer sheath proximate to the collar only partially contracts, allowing the tube to be extended for an additional 2 nm in the final state. The authors concluded that the thicker peptidoglycan layer prevented the continuous contraction of the tail, thus it is necessary to pause briefly, allowing the hydrolase domain of the hub-hydrolase to continue the catalytic action, for completely degrading the thick peptidoglycan layer. However, the particles of the transitional state and the final state are both in solution, and there is no condition that hinders the contraction. What makes the contraction stay at the first four layers but not be further transmitted. Is it possible that after the assembly is completed, there are three kinds of particles: one is relatively stable (pre-contraction state), the other two are relatively unstable which are more likely to contract during the purification process. The so-called transitional state is indeed another final state in which the first four layers are not prone to contracting?

2) Line 119-139: What is the local resolution of the hub-hydrolase, the Extended Data Fig. 6a and b show that it is low resolution. This part should be stuck very tight, thus what causes the lower resolution of this part? And neither the figures nor the movies show the electron microscopy density of the end of the tube in the post-contraction state, are hub hydrolases and tube tails present, and are they still attached to the tube in the final state? A better map of this section will help a lot to improve the quality of the manuscript.

3) Line 114-115: Fig2b shows that the diffocin spike carries ferric ions at the tip. How the presence of ferric ions was determined, whether by electron microscopy density or other methods, and whether the histidine doublets are conserved in the contraction injection system?

4) Line 200-212: The diameter of L32 in the transitional state is almost identical to that in the pre-contraction state. Fig 5a and 5b show that the N-terminal of TMP is located between the collar and the inner tube of L32 in the pre-contraction state. Fig 4b shows slight changes on the outside, but the inner tube and upper collar don't seem to change much. If the inner side of L32 where the TMP is located does not change, how does the TMP leave?

5) Line 221-225: What is the local resolution of the TMP density? What is the difference in electron microscopy density of C6, C3 and C1 symmetries? Is it possible for the electron density map of C6 to fit a six-helix bundle? Please provide the above map and add the TMP processing process to the flowchart.

Minor points:

Line 21: "antibiotic refractory recurrent disease" should be "antibiotic-refractory recurrent disease"

Line 55: "its method infiltrating" should be "its method of infiltrating"

Line 120: "a conserved gp27-like unifunctional hub domain", gp27 should be introduced first.

Line 146: "TRI-1" should be "Tri1"

Line 188: Extended Data Fig. 7d should be Extended Data Fig. 7c?

Line252: "highlight" should be "highlights"

Lin416: "unique asymmetrical unit" should be "unique asymmetric unit"

Lin538: in Fig 3a and 3b, the layers are not clearly shown.

Reviewer #2:

Remarks to the Author:

This paper investigates the pre-contraction and post-contraction states of a tailocin that targets Gram-positive bacteria using cryoEM. Detailed analyses reveal the mechanism by which the bacteriocin is able to contract and penetrate the bacterial cell wall, as well as how it might form a pore in the cell membrane that kills the target cell. This is an interesting study that is very clearly explained by the authors and contains excellent illustrations.

From this paper it is not clear how the contraction is triggered, as the receptor binding protein is not part of the analysis. How does the interaction between the bacterial receptor and tailocin induce a conformational change that results in the end the lowest energy post-contraction state of the tailocin? Does the fact that pre- and post-contraction diffocin particles were present in the purified sample shed any light on what factor(s) triggers these contractions?

Minor comments:

- It would be nice if the authors briefly mention that non-contractile tailocins (F-type) also exist.
- Why did the authors decide to use a hybrid diffocin gene cluster for their study? Should it matter that this engineered tailocin targets specifically *Clostridioides difficile* when the purpose was to understand tailocins that target Gram-positive bacteria?
- Lines 76-77: "With these maps, we were able to assign 10 of the 15 diffocin genes.." should be "With these maps, we were able to assign 10 of the 15 diffocin gene products....".
- There seems to be no discussion in the text of Extended Data Fig. 6.

Reviewer #3:

Remarks to the Author:

This is a beautifully executed and well documented study reporting the first structure of a Gram-positive-targeting contractile R-type pyocin, diffocin, targeting the medically important opportunistic pathogen *Clostridium difficile*. Despite the complexity of the system, the quality of the illustrations and text bring a very clear and insightful description that is easy to follow and a pleasure to read. The study documents the intricacies as well as commonalities of Gram-positives versus Gram-negatives targeting R-type pyocins, in particular illustrating the adaptations required to penetrate the different cell envelope compositions.

This study will be of high interest to the broad community studying tailed phages and bacterial contractile injection systems, and will form a landmark study to those seeking to engineer CIS into next generation antimicrobials.

I have a few minor points for the authors to consider.

- The authors find two alternative conformations for the post-contraction state of diffocin, and propose one represents a transitional state and the other a final state. Is the number of particles following 3D classification (15,513 / 18,509) representative for the ratio of the two conformational states, or a merely subset selected for 3D reconstruction? This ratio and the fact that the transition zone is not stochastically distributed along the length of the sheath amongst different particles suggest this 'transitional state' is a longer lived state, possibly a conformationally trapped, alternative end state in the colar region. If the observed 'transition state' particles are still en route

to the final state, what would make the 'transitional state' stall and what would make it progress?

- The reconstruction and illustrations of the post-contraction states focus on the collar-to-baseplate (i.e. contracted sheath) region of the particle. It would be good to include a description of the tube & spike region extending out from the base plate, i.e. what is the average length from spike to base plate and how does this compare with the average height of the C. diff cell envelope?

- Ln 114-18 The tip of the diffocin spike is said to coordinate a ferric iron. How was this density assigned as being ferric iron?

Responses to reviewers' comments

Summary of responses and revision:

We thank the three reviewers for the careful review and support of our paper! As you can see from the itemized responses below, we have addressed the reviewers' comments fully and revised the manuscript accordingly. For Reviewer #1 and Reviewer #3's question about the status of the spike-proximal tube region after contraction, we have carried out extensive data processing to classify this region of the post-contraction diffocin to answer this question. For the convenience of perusal, we have copied the reviewers' comments in **black** and our responses are shown in **blue**. The line numbers of changed text in the revised manuscript are also indicated for your ease of navigation. The corresponding changed text is highlighted in **blue** in the revised manuscript.

Reviewer #1 (Remarks to the Author)

In this manuscript, Cai et al. reported the cryo-EM structures of the diffocin from bactericidal contractile system against Gram-positive bacteria in its pre- and post-contraction states, enriching our understanding on the contraction injection system. Furthermore, they found that the diffocin harbors a multifunctional hub-hydrolase protein that connects the tube and baseplate for the degradation of peptidoglycan during penetration. Additionally, the full-length tape measure protein forms a coiled-coil helix bundle homotrimer spanning the entire length of the diffocin. The following major concerns are raised for the authors to revise and improve the manuscript.

1) Line 264-273: In the transitional state, the four-layer sheath proximate to the collar only partially contracts, allowing the tube to be extended for an additional 2 nm in the final state. The authors concluded that the thicker peptidoglycan layer prevented the continuous contraction of the tail thus it is necessary to pause briefly, allowing the hydrolase domain of the hub-hydrolase to continue the catalytic action, for completely degrading the thick peptidoglycan layer. However, the particles of the transitional state and the final state are both in solution, and there is no condition that hinders the contraction. What makes the contraction stay at the first four layers but not be further transmitted. Is it possible that after the assembly is completed, there are three kinds of particles: one is relatively stable (pre-contraction state), the other two are relatively unstable which are more likely to contract during the purification process. The so-called transitional state is indeed another final state in which the first four layers are not prone to contracting?

Response: This is an excellent point. In Supplementary Fig. 1a and Fig. 6b-d, we captured two types of post-contraction collar (post-collar) particles: "post-collar long", where the sheath is not fully contracted, and "post-collar short", where the sheath is fully contracted. There is only one interface between the collar and sheath in "post-collar long", whereas there are two interfaces in "post-collar short", indicating a more stable state (Fig. 6f-h). Given that the pre-contraction state is metastable and the post-contraction state is stable (Caspar, D. Biophys J, 1980), and that "post-collar short" has two interfaces, we designate that "post-collar short" to be a stable, **final** state and "post-collar long" to be a relatively metastable, **transitional** state. We have revised the paper to clarify about our rationale on assigning transitional and final states, see lines 210-212 and 219-226.

2) Line 119-139: (1) What is the local resolution of the hub-hydrolase, the Extended Data Fig. 6a and b show that it is low resolution. This part should be stuck very tight, thus what causes the lower resolution of this part? (2) And neither the figures nor the movies show the electron microscopy density of the end of the tube in the post-contraction state, are hub hydrolases and tube tails present, and are they still attached to the tube in the final state? A better map of this section will help a lot to improve the quality of the manuscript.

Response: These two questions are addressed separately below.

(1) The local resolution of the hub-hydrolase was added in Supplementary Fig. 5g. The local resolution of the hydrolase domain of hub-hydrolase (CD1368) is ~4-5 Å, lower than that of the hub domain (3 Å resolution). We believe this is because the baseplate wedge of diffocin (with a diameter of 24 nm) is less compacted than the baseplate wedge of pyocin (with a diameter of 20 nm), and the hydrolase domain lacks specific interactions with the baseplate wedge. Both factors increase the local motion of the peripheral region of the diffocin baseplate and result in lower local resolution.

(2) We have added a representative cryoEM image, 2D classification and 3D refinement density of the end of the tube in the post-contraction state in Supplementary Fig. 5h-j as suggested. In the 2D classification and 3D refinement results, 74% of particles present as a tube, with no significant density of the spike or hub-hydrolase due to the limited number of particles. Meanwhile, 26% of particles present as a baseplate without the sheath. These baseplates may have detached from the sheath and did not contract in a small number of cases.

3) Line 114-115: Fig2b shows that the diffocin spike carries ferric ions at the tip. How the presence of ferric ions was determined, whether by electron microscopy density or other methods, and whether the histidine doublets are conserved in the contraction injection system?

Response: We interpret these densities as ferric ions because the tip structure of the spike has three sets of histidine doublets, similar to the spikes in R-type pyocin, P2 phage and phi92 phage. In the crystal structure of these spikes, a ferric ion coordinated by three sets of histidine doublets is located at the very tip (Ge, P. Nature, 2020; Browning, C. Structure, 2012). These histidine doublets are conserved in the contraction injection system, forming a structure that stabilizes the tip for membrane penetration. Nonetheless, we have changed the statement by referring this density as “putative ferric iron” in the revised manuscript. See lines 118-120.

4) Line 200-212: The diameter of L32 in the transitional state is almost identical to that in the pre-contraction state. Fig 5a and 5b show that the N-terminal of TMP is located between the collar and the inner tube of L32 in the pre-contraction state. Fig 4b shows slight changes on the outside, but the inner tube and upper collar don't seem to change much. If the inner side of L32 where the TMP is located does not change, how does the TMP leave?

Response: We are not certain on how the TMP leaves but could offer the following speculation.

Fig. 7e-g demonstrate side-chain interactions between the C-termini of the TMP and the N-termini of the spike; such interactions are less prevalent in the N-termini of the TMP with the collar/L32. The resolution difference between TMP-spike and TMP-collar/L32 densities indicates the interaction of TMP-spike is more rigid (Supplementary Fig. 8c). When the spike detaches from the tube after R-type bacteriocin contraction (Scholl, D. Annu. Rev. Virol., 2017), the TMP trimer may leave while still interacting with the spike.

It is also possible that the TMP may not follow the spike. The TMP could still temporarily reside in the inner tube and finally leave due to entropic favorability. AlphaFold predictions of the TMP trimer indicate a predicted globular native conformation of the TMP trimer under standard biological conditions, as opposed to the linearized form fashioned before ejection due to biochemical constraints from the neighboring inner tube. This indicates that it is likely entropically favorable for the individual monomers of the TMP to go from a linear metastable state in the tube to a globular stable state upon ejection into the extracellular region, driving ejection from the tube.

5) Line 221-225: (1) What is the local resolution of the TMP density? (2) What is the difference in electron microscopy density of C6, C3 and C1 symmetries? (3) Is it possible for the electron density map of C6 to fit a six-helix bundle? (4) Please provide the above map and add the TMP processing process to the flowchart.

Response: These four questions are addressed separately below.

(1) The local resolution of the TMP density is ~3-6 Å, with the highest resolution observed in the C-terminus of TMP where it interacts with the spike.

(2) We divided the entire diffocin into six segments for data processing of TMP density. Taking the TMP density of the pre-collar segment as an example, the density in C6 symmetry presents as a hollow cylinder without any secondary structure features. After relaxing the symmetry to C3 and performing alignment-free 3D classifications using spherical masks covering the tape measure protein regions, the map of the TMP presents as three stick-like densities. This feature was further confirmed after relaxing the symmetry to C1.

(3) No, a six-helix bundle cannot be fitted in the TMP density with C6 symmetry.

(4) We added the TMP process workflow, which includes the density in C6, C3 and C1 symmetries, and the local resolution of the TMP density in Supplementary Fig. 8.

Minor points:

Line 21: “antibiotic refractory recurrent disease” should be “antibiotic-refractory recurrent disease”

Response: Fixed. New line: 21

Line 55: “its method infiltrating” should be “its method of infiltrating”

Response: Fixed. New line: 57

Line 120: “a conserved gp27-like unfunctional hub domain”, gp27 should be introduced first.

Response: Fixed. New line: 125

Line 146: “TRI-1” should be “Tri1”

Response: Fixed. New line: 155

Line 188: Extended Data Fig. 7d should be Extended Data Fig. 7c?

Response: Great catch! Fixed. New line: 197

Line252: “highlight” should be “highlights”

Response: Fixed. New line: 267

Lin416: “unique asymmetrical unit” should be “unique asymmetric unit”

Response: Fixed. New line: 432

Lin538: in Fig 3a and 3b, the layers are not clearly shown.

Response: We have modified the color of different layers of sheath in Fig 5a and b.

Reviewer #2 (Remarks to the Author)

This paper investigates the pre-contraction and post-contraction states of a tailocin that targets Gram-positive bacteria using cryoEM. Detailed analyses reveal the mechanism by which the bacteriocin is able to contract and penetrate the bacterial cell wall, as well as how it might form a pore in the cell membrane that kills the target cell. This is an interesting study that is very clearly explained by the authors and contains excellent illustrations.

From this paper it is not clear how the contraction is triggered, as the receptor binding protein is not part of the analysis. How does the interaction between the bacterial receptor and tailocin induce a conformational change that results in the end the lowest energy post-contraction state of the tailocin? Does the fact that pre- and post-contraction diffocin particles were present in the purified sample shed any light on what factor(s) triggers these contractions?

Response: The contraction can be triggered by physiologically relevant events such as binding to receptors on the bacteria surface, or by non-physiological disturbances such as mechanical stress in the process of diffocin purification. From the structural data of both R-type diffocins and pyocins, it seems that the protein-protein interactions between the baseplate triplex are primarily responsible for keeping the particle in the uncontracted state. When (we believe six) RBP interactions occur on the surface of the cell, the mechanical force is enough to break the baseplate protein interactions such that the entire ring changes conformation, resulting in contraction.

In solution, anything that could disrupt protein-protein interactions (changes in pH, water activity, temperature, freezing) could result in spontaneous breaking of enough of the baseplate interactions to result in the same irreversible baseplate contraction, resulting in accumulation of contracted particles.

Minor comments:

- It would be nice if the authors briefly mention that non-contractile tailocins (F-type) also exist.

Response: Indeed, F-type tailocins, such as F-type pyocins and monocins, are also very important phage tail-like bactericidal protein complexes. This is now indicated in lines 23-25.

- Why did the authors decide to use a hybrid diffocin gene cluster for their study? Should it matter that this engineered tailocin targets specifically *Clostridioides difficile* when the purpose was to understand tailocins that target Gram-positive bacteria?

Response: The diffocin particles we imaged has been subject to *in vivo* validation as a precision antimicrobial agent (Gebhart, D. mBio, 2015) and will hopefully find eventual use in humans to treat or prevent *C. difficile* colitis. We are hopeful that our results will be useful for structure guided engineering of a human therapeutic that could save lives. The diffocin was directly derived from a naturally occurring *C. difficile* tailocin, and the only changes made involved swapping the RBP and associated chaperone from a highly related phage. This swap **broadens** the host range of the original tailocin. All other components involved in maintaining pre- or post-contraction states, triggering contraction, cell wall hydrolysis and surface penetration, etc. remain unchanged. The bactericidal activity of these hybrid particles mirrors its wild type counterpart, and there is nothing to suggest that the mechanism of contraction is any different. On a practical note, we find that the hybrid diffocin expresses more efficiently and gives higher yields in our *Bacillus*-based production system.

- Lines 76-77: "With these maps, we were able to assign 10 of the 15 diffocin genes.." should be "With these maps, we were able to assign 10 of the 15 diffocin gene products....".

Response: Corrected. See line 80 in the revised paper.

- There seems to be no discussion in the text of Extended Data Fig. 6.

Response: This is now cited in lines 268-274.

Reviewer #3 (Remarks to the Author):

This is a beautifully executed and well documented study reporting the first structure of a Gram-positive-targeting contractile R-type pyocin, diffocin, targeting the medically important opportunistic pathogen *Clostridium difficile*. Despite the complexity of the system, the quality of the illustrations and text bring a very clear and insightful description that is easy to follow and a pleasure to read. The study documents the intricacies as well as commonalities of Gram-positives versus Gram-negatives targeting R-type pyocins, in particular illustrating the adaptations required to penetrate the different cell envelope compositions.

This study will be of high interest to the broad community studying tailed phages and bacterial contractile injection systems, and will form a landmark study to those seeking to engineer CIS into next generation antimicrobials.

Response: Thank you for your supportive comments!

I have a few minor points for the authors to consider.

- The authors find two alternative conformations for the post-contraction state of diffocin, and propose one represents a transitional state and the other a final state. (1) Is the number of particles following 3D classification (15,513 / 18,509) representative for the ratio of the two conformational states, or a merely subset selected for 3D reconstruction? (2) This ratio and the fact that the transition zone is not stochastically distributed along the length of the sheath amongst different particles suggest this 'transitional state' is a longer lived state, possibly a conformationally trapped, alternative end state in the collar region. (3) If the observed 'transition state' particles are still en route to the final state, what would make the 'transitional state' stall and what would make it progress?

Response: These three questions are addressed separately below.

(1) The number of particles of the 3D classification (15,513 / 18,509) is representative of the ratio between the transitional and final states.

(2) We do not believe the transitional state is another final state based on the particle ratio alone. The number of pre-contraction diffocin particles is over four times that of the post-contraction diffocin particles (Supplementary Fig. 1a). While it is a long-lived state in solution, it is a metastable state (Caspar, D. Biophys J, 1980).

(3) While we are not certain why the transitional state stalls, we speculate that the transitional state stall may be needed for hydrolase-catalyzed degradation of the thick peptidoglycan layer of *C. difficile*. We are not sure what triggers further contraction.

- The reconstruction and illustrations of the post-contraction states focus on the collar-to-baseplate (i.e. contracted sheath) region of the particle. It would be good to include a description of the tube & spike region extending out from the base plate, i.e. what is the average length from spike to base plate and how does this compare with the average height of the *C. diff* cell envelope?

Response: We have carried out data processing on the spike region in the post-contraction diffocin and the results are added in Supplementary Fig. 5h-j. Many post-contraction diffocin particles had spike-proximal regions located at the edge of holes of Quantifoil grid (indicated by red arrows in the Response Fig. 1 below). Consequently, we do not have enough analyzable particles of the post-contraction tube-spike region, resulting in limited resolution. Although there is no 3D density map to measure the length from the spike to baseplate, we measured the length in the cryoEM images. The average length is about 50 nm. This is close to the 47 nm obtained by subtracting the length of the post-contraction diffocin from the collar to baseplate (99 nm) from the length of the pre-contraction diffocin from the collar to spike (146 nm), as described in the Results section (lines 70-71) and Supplementary Fig. 10c. This length is shorter than ~58 nm average height of the *C. diff* cell envelope.

Response Fig. 1 | CryoEM images of diffocin particles. Red arrows indicate representative post-contraction diffocin particles with spike-proximal regions located at the edges of holes of the Quantifoil grid. Scale bar, 100 nm.

- Ln 114-18 The tip of the diffocin spike is said to coordinate a ferric iron. How was this density assigned as being ferric iron?

Response: Please see response above to a similar question, question #3 from Reviewer #1. We have changed “ferric iron” to “putative ferric iron”, as although the coordination of the density is the same, we do not have definitive proof of its identity. See lines 118-120 in the revised paper.

In **summary**, we thank three reviewers for the careful reviews and valuable suggestions! We hope you would agree that we have addressed all the concerns raised by performing additional computational analyses and adding new panels, leading to an improved paper.

Reviewers' Comments:

Reviewer #1:

Remarks to the Author:

The quality of the manuscript has been greatly improved after revision. All my concerns have been properly addressed.

Reviewer #2:

Remarks to the Author:

The authors have addressed the various issues that were raised and made corrections to the manuscript when needed. I have no further suggestions.

Reviewer #3:

Remarks to the Author:

I had few comments, which the authors satisfactorily addressed in their rebuttal and revised manuscript. This is an excellent addition to our understanding of R-type pyocins.